# Underwater acoustic target recognition method based on a joint neural network

**Xing Cheng Han**[1,2]*, **Chenxi Ren**[1,2], **Liming Wang**[1,2], **Yunjiao Bai**[3]

**1** State Key Laboratory of Dynamic Testing Technology, North University of China, Taiyuan, China, **2** School of Information and Communication Engineering, North University of China, Taiyuan, China, **3** Department of Mechanics, Jinzhong University, Jinzhong, China

* zbhanxc@nuc.edu.cn

## Abstract

To improve the recognition accuracy of underwater acoustic targets by artificial neural network, this study presents a new recognition method that integrates a one-dimensional convolutional neural network and a long short-term memory network. This new network framework is constructed and applied to underwater acoustic target recognition for the first time. Ship acoustic data are used as input to evaluate the network performance. A visual analysis of the recognition results is performed. The results show that this method can realize the recognition and classification of underwater acoustic targets. Compared with a single neural network, the relevant indices, such as the recognition accuracy of the joint network are considerably higher. This provides a new direction for the application of deep learning in the field of underwater acoustic target recognition.

**Data Availability Statement:** The ship radiated data used in this paper came from the ShipsEar dataset recorded in different areas of the Spanish coast between 2012 and 2013. The website is https://atlanttic.uvigo.es/underwaternoise/.

## 1. Introduction

In recent years, with the development of science and technology, underwater acoustic target recognition technology has attracted increasing attention from scientific and technical personnel because it is a vital issue in the field of underwater acoustic signal processing.

The problem of underwater acoustic target recognition is extremely complex. The multifaceted underwater environment causes distortion in radiated noise [1], making underwater acoustic target recognition more difficult than conventional speech recognition. As technological development is slow, more accurate underwater acoustic target recognition methods must be investigated.

The task of underwater acoustic target recognition is to analyze underwater acoustic signals received by a sonar system and extract the features of targets. At present, deep learning is a popular technology in various industries. Owing to its very strong feature extraction and optimization capabilities, deep learning has opened up a new development direction for underwater acoustic target recognition technology [2–9]. Many researchers apply convolutional neural network (CNNs) to underwater acoustic target recognition [10–13].

The long short-term memory (LSTM) architecture is suitable for processing and forecasting events with long intervals in time series. The analysis of ship-radiated noise depends largely on

**Funding:** The manuscript is funded by:(1)Science and Technology Innovation Project of Colleges and Universities in Shanxi Province(China), the award number is 2020L0301;(2)Science and Technology Innovation Project of Colleges and Universities in Shanxi Province(China), the award number is 2020L0595;(3)Fundamental Research Program of Shanxi Province(China), the award number is 20210302124545.

**Competing interests:** The authors have declared that no competing interests exist.

local time-frequency information and time-series related information; therefore, LSTM can be utilized for underwater acoustic target recognition [14–17].

Because the recognition framework of a single neural network makes the extraction of all features of underwater acoustic signals challenging [18–20], the research is usually focused on the development of deeper and more complex networks [21–29], which however are more difficult to train (in terms of training data size and labeling requirements). Therefore, building a new network model by combining various network structures may be a good solution. Studies on joint neural networks are mostly based on traditional two-dimensional (2D) CNN and LSTM. However, owing to the characteristics of the network model, 2D CNN seems to perform better in the field of image recognition. Conversely, 1D CNN are usually employed in speech-processing fields such as sequence modeling and natural language processing, where the use of 1D CNN reduces the amount of computation required. Therefore, in this study, a 1D CNN and LSTM network model are combined to identify underwater acoustic targets, with the aim to obtain a network with a higher training speed and recognition rate.

Based on the related research on the acoustic target recognition technology in this study, a network recognition framework is built and a new type of neural network is established by combining the advantages of the 1D CNN and the LSTM. The network is trained using the extracted characteristics of ship acoustic signals as input.

## 2. Recognition principle

### 2.1 Convolutional neural network

CNNs usually include three types of network layers: convolutional, pooling, and fully connected. The pooling layer is also called the down-sampling layer. The convolution and pooling layers usually contain multiple feature matrices, which are generated by different convolution cores. Dimension reduction of data can be achieved through multiple convolution and pooling layers. Finally, the predicted category labels can be obtained through the fully connected output layer.

As the task to be completed in this study is to recognize one-dimensional underwater acoustic signals, a one-dimensional convolutional neural network (1D CNN) is used [11, 12]. 1D CNN model is shown in Fig 1.

The difference between convolutional neural networks and other networks is convolution, and it is the most critical operation for convolutional neural networks. Through the convolution kernel, the convolution layer can extract important features from inputs and form feature vectors. Its operational expression is

$$X^l = X^{l-1} * W^l + b^l \tag{1}$$

where $X^l$ and $X^{l-1}$ are the eigenvectors of the $l$ and the $l-1$ layers respectively, $W^l$ is the convolution kernel, $b^l$ is a biased vector, and $*$ is the convolution operation.

Discriminative features are extracted from input data through the linear transformation of the convolution operation, and then the characteristics more suitable for classification are obtained through the nonlinear transformation of the activation operation. The activation operation must be completed by setting the activation function. In this study, a common rectified linear unit (ReLU) activation function was used, and its expression is

$$f(x) = max\left(0, x_j^l\right) \tag{2}$$

where x = input value. Owing to the extraction of characteristics of input data with high dimensions, it is easy to cause overfitting of the neural network to the training dataset.

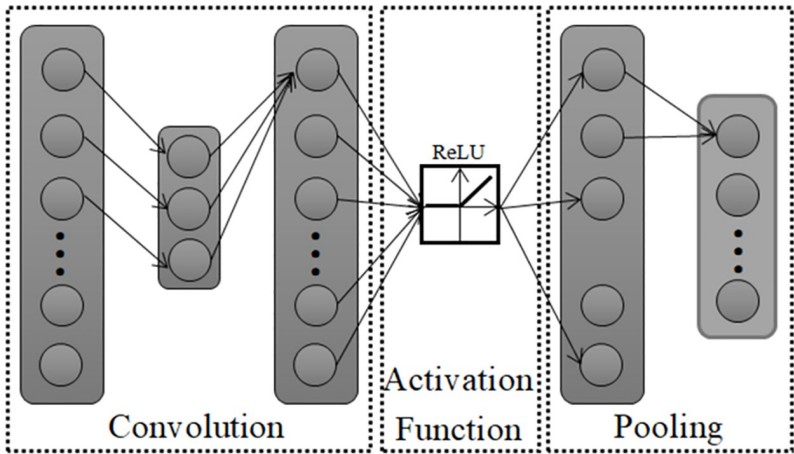

**Fig 1. 1D-Convolutional neural network model.**

Therefore, usually pooling layers are joined to improve the operation speed, reduce the training time, and effectively prevent training data overfitting [7]. The pooling layers are calculated by sliding the kernel on the input matrix. However, the operation of the pooling kernel does not contain any parameters. Therefore, the pooling layers are usually divided into maximum pooling and average pooling. The maximum or average values of the matrix elements in the specified range of the previous layer are taken as the output of this layer. The output of the pooling layer is

$$X^l = S[X^{l-1}] \qquad (3)$$

where $S$ is the down-sampling rule, Maximum pooling is used in this study, and the maximum pooling expression is

$$S[X^{l-1}] = \max_{0<n\leq w} x_n^{l-1} \qquad (4)$$

where $x_n^{l-1}$ represents the nth neuron in the eigenvector output by convolutional layer $l-1$, and $w$ is the pool size. After multiple convolution and pooling layers, the classification layer can be used to complete the classification and recognition tasks.

## 2.2 Long short-term memory

Because the LSTM network can analyze and extract data from each sequence, it is often widely used to process sequence data and model short-term or long-term dependencies between data [8]; therefore, the LSTM model is used in this study.

It is a network structure with cyclic links connected to each other. As a whole, the LSTM network is still a recurrent neural network, but there are small loops of LSTM blocks in the network. The difference between the network and ordinary recurrent neural network is that neurons are replaced with LSTM blocks. Its biggest advantage is that it can link multiple nodes, connect the nodes of the same hidden layer in series, and realize parameter sharing among all nodes, making it completely different from other networks in technology [9]. The LSTM block is shown in Fig 2.

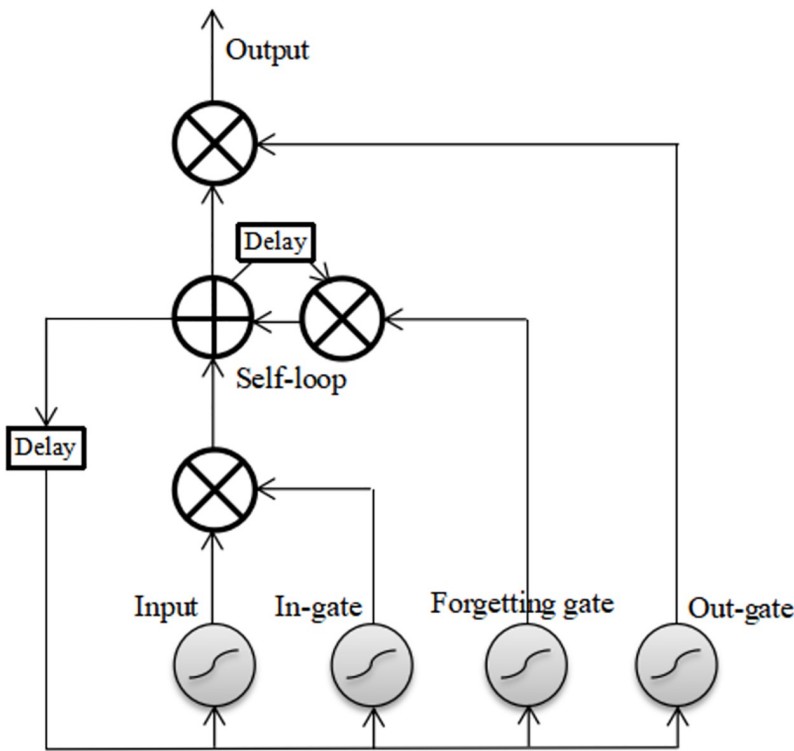

**Fig 2. LSTM block model diagram.**

The update calculation of forget gate implementation is as follows:

$$f_{gate} = \sigma\left(W_f \cdot \left[a^{(t-1)}, x^{(t)}\right] + b_f\right) \tag{5}$$

where $W_f$ is the weight, $a^{(t-1)}$ is the input of the previous cell, $x^{(t)}$ is the input of the current cell, $\sigma$ is the sigmoid function, and $b_f$ is the bias. The forget gate reads $a^{(t-1)}$ and $x^{(t)}$ and then outputs a value between 0 and 1 to the cell state $c^{(t-1)}$, where 1 indicates that information is completely retained and 0 indicates that information is completely dropped.

The in gate is used to decide how much new information is added to the current cell state, and the specific process of the in gate is expressed as

$$i_{gate} = \sigma\left(W_i \cdot \left[a^{(t-1)}, x^{(t)}\right] + b_i\right) \tag{6}$$

$$C_{tanh} = \tanh\left(W_C \cdot \left[a^{(t-1)}, x^{(t)}\right] + b_C\right) \tag{7}$$

$$C^{(t)} = f_{gate} * C^{(t-1)} + i_{gate} * C_{tanh} \tag{8}$$

where $W_i$ and $W_c$ are the weight, $b_i$ and $b_c$ are the bias, the update gate first uses the sigmoid function to calculate the information to be updated, and then uses the tanh function to extract the updated content.

The output gate was used to determine the final output information of the cells. First, the sigmoid function calculates which information needs to be output, and then the tanh layer is

used to output this information. The specific calculation process is as follows:

$$O_{gate} = \sigma\left(W_O \cdot \left[a^{(t-1)}, x^{(t)}\right] + b_O\right) \tag{9}$$

$$a^{(t)} = O_{gate} * \tanh\left(C^{(t)}\right) \tag{10}$$

where $W_o$ is the weight, and $b_o$ is the bias. The computation of the LSTM network is more complicated than that of an ordinary recurrent neural network, but its performance in learning long-term dependence is better than that of any known cyclic network, and it performs well in sequential processing tasks.

## 3. Experimental data

### 3.1 Dataset

The ship-radiated noise data used in this study came from the ShipsEar [30] dataset recorded in different areas of the Spanish coast between 2012 and 2013. This dataset consists of 90 acoustic records of 11 types of ships and environmental noise within 15 seconds– 10 minutes. According to the annotation in the original dataset, they can be grouped into four categories based on the types of ships, namely A, B, C, and D, and E for environmental noise. The types of ships included in each category are listed in Table 1.

### 3.2 Data processing

As the original data sets are all real data collected from the ocean, there are some problems such as excessive noise and blank segments in some datasets; therefore, the dataset of 90 acoustic signals needs to be preprocessed.

First, a part of the acoustic signals with poor collection effect was removed. In the remaining acoustic signals, the blank segment left during the collection was removed manually, and the acoustic signals was de-noised. Some acoustic signals with low sounds were enhanced. To enlarge the dataset, we split the original 90 acoustic signals into 3seconds fragments.

To characterize the features of the acoustic signals more comprehensively, we extracted as many features as possible for feature fusion as the network input, to achieve a better recognition effect. In addition to the traditional features like Mel-spectrogram and Mel-Frequency Cepstral Coefficients, we also used three features that are often used in music theory, namely chromatogram, spectral contrast and tonnetz, the following will be introduced separately:

The first is to extract the Mel-spectrogram [31], obtain the Mel Bank Features based on Mel-scale, and the length of Mel spectrum is set as 128. Then the columns of the resulting matrix are compressed, the average value of each row is calculated, and an eigenvector of (128,1) is returned.

The second is to extract the mel-frequency cepstral coefficients [32]. It is a kind of coefficient obtained by utilizing the human nonlinear auditory system, performing nonlinear

**Table 1. Dataset classification.**

| Category | Ship types |
|---|---|
| A | Fishing boats; Trawlers; Mussel boats; Tugboats; Dredgers |
| B | Motorboats; Pilot boats; Sailboats |
| C | Passenger ferries |
| D | Ocean liner; Ro-Ro vessels |
| E | Background noise recordings |

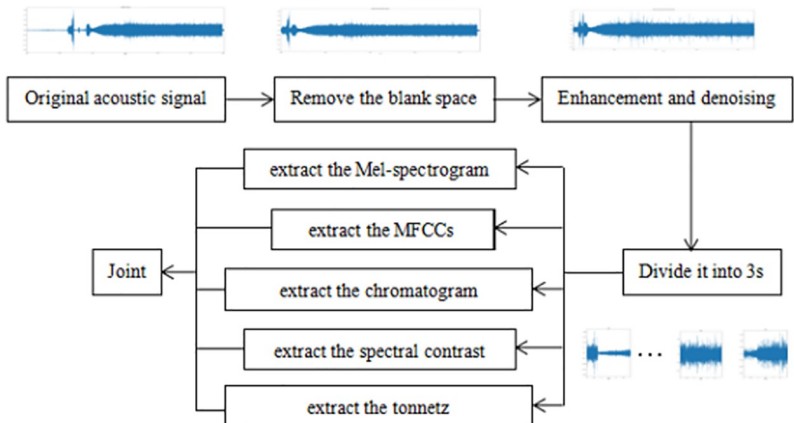

**Fig 3. Data processing flow chart.**

conversion to the acoustic signal frequency spectrum corresponding to the Mel-spectrum, and then transforming to cepstrum. Here, the row dimension of its output is set to 40, and then column compression is performed on the obtained coefficient matrix to obtain the eigenvector with the final dimension of (40,1).

The third is to calculate the chromatogram from the results of the short-time Fourier transform of the acoustic signal [33]. Because the feature it reflects is related to twelve different pitch levels, the resulting vector size is (12,1).

Fourth, spectral contrast is extracted, and spectral contrast based on the octave scale can be used to extract the relative spectral characteristics of acoustic signal [34]. Through this step, the eigenvector size obtained is (6, 1).

Fifth, tonnetz is extracted, which is mainly used to analyze the chord relationship of sound [35]. In this step, the pure fifth, third, and minor third are used as two-dimensional coordinates to obtain the feature vectors of (6,1).

After the five features are extracted, the feature vectors obtained are fused, and for each acoustic signal, a feature vector with a dimension of (192,1) is provided as the input of the network. The processing flow chart is shown in Fig 3.

To make the five fusion extraction features of the input acoustic signals express more comprehensively, we conducted t-SNE visualization of single extracted Mel spectrum, MFCCs feature and fusion feature, and the results are shown in Fig 4.

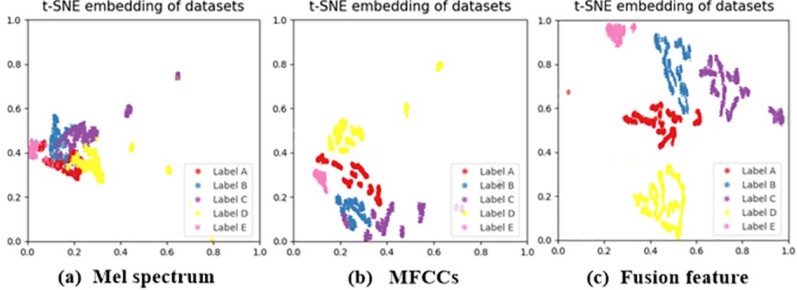

**Fig 4. t-SNE visualization result of the Mel spectrum, MFCCs feature and fusion feature.** (a) Mel spectrum; (b) MFCCs; (c) fusion feature.

**Table 2. Actual size of the dataset used.**

| Category | Acoustic signal serial number | The number of data | Total |
|---|---|---|---|
| A | 13,15,28, 46–49,66,73–76,80, 93–96 | 1040 | 4900 |
| B | 26,27,29,30,33,50–52,56,57,68, 70,72,77,79 | 790 | |
| C | 6,10,40,42,43,52–54,59–65,67 | 1340 | |
| D | 18–20,22,24,25,58,69,71,78 | 1135 | |
| E | 81–92 | 595 | |

It can be seen from the above figures that the fusion feature is more separable than the single feature, mainly because the spectral contrast and other musical theory features used can capture the tonal features of the acoustic signals more sensitively. Therefore, the subsequent research in this paper will take fusion features as input.

We manually screened the acoustic signals in accordance with the original annotation, removed some unprocessed acoustic signals with poor recording effect, and processed the remaining acoustic signals to obtain the actual data set.

To make it easier for other researchers to use the ShipsEar dataset, each acoustic signal piece in the dataset is assigned a number, and the serial number used is indicated in Table 2.

To better verify the network, 4900 samples were randomly selected and divided into a training set and a test set in a ratio of four to one. The number of samples was 3920 for the training set and 980 for the test set.

### 3.3 Network construction

The1D-CNN network uses one-dimensional convolution to process a one-dimensional sequence model, which is widely used in acoustic signal recognition. Because the ship's voyage is a continuous process, its acoustic signal characteristics must have continuity in time, so we can consider the method of processing time series signals to identify the ship target.

The characteristics of the ship's underwater acoustic signal is time-varying, and we can use the LSTM network to capture the characteristics of the current moment and the historical information of the previous moment. Combined with a one-dimensional CNN and LSTM network, the system can quickly adapt to signal changes and improve the recognition accuracy.

Therefore, we build a joint model of the1D-CNN and LSTM network. The 1D CNN part of the network consists of two convolution layers and two pooling layers alternately. The pooling layer adopts maximum pooling, followed by a dropout layer, and the LSTM part consists of one LSTM layer and one dropout layer. Finally, it is sent into the dense layer for the classified output, and the network model is shown in Fig 5.

Specific parameters of the network are shown in Table 3:

## 4. Experimental results

### 4.1 Training network

Based on the aforementioned dataset, we randomly divided all 4900 acoustic signal clips into training and test sets, and the test set accounted for 20% of the total data. After setting up the joint network model, we set the network training parameters as shown in Table 4:

### 4.2 Training results

After 100 epochs, we obtained the loss and accuracy curves, as shown in Figs 6 and 7. The curve composed of blue points is the change curve of the training set, and the red curve is the test set.

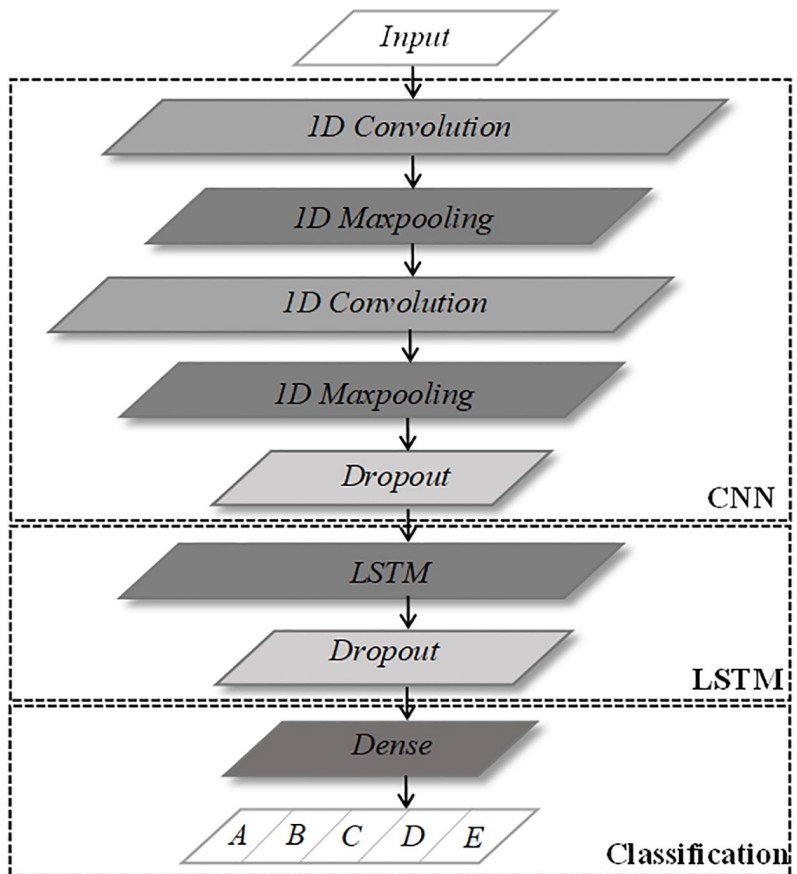

**Fig 5. Joint network model.**

The classification accuracy of the joint network for the data set reached 96.73% in the training set and 92.14% in the test set.

In order to verify the performance of the joint network proposed in this study, we compared its recognition accuracy with 1D-CNN and LSTM networks, and the results are shown in Table 5.

By comparison, we found that the recognition accuracy of the joint network was 14.46% higher than that of the LSTM network in the training set, 10.75% higher than that of the 1D-CNN, and 16.04% higher than that of the LSTM network in the test set, 7.96% higher than that of the 1D-CNN network.

To intuitively see the recognition performance of the three networks on the ShipsEar dataset, we visualized the recognition results on the test set by drawing the confusion matrix, and the results are shown in Fig 8.

In the figure, 0 to 4 of the horizontal and vertical coordinates represent labels A to E. By using the confusion matrix, we can calculate the recognition accuracy of the three networks for the five types of ship targets, as shown in Table 6.

The recognition accuracy of the joint network for the five types of targets is the highest among the three types of networks. Therefore, we can deduce that the joint network is of considerable help in improving the accuracy of underwater acoustic target recognition.

**Table 3. Network parameter table.**

| Layer | Output Shape | Param |
|---|---|---|
| Conv_1D | 191×64 | 256 |
| Maxpooling1D | 63×64 | 0 |
| Conv_1D | 62×128 | 24704 |
| Maxpooling1D | 20×128 | 0 |
| Dropout | 20×128 | 0 |
| LSTM | 32×1 | 20608 |
| Dropout | 32×1 | 0 |
| Dense | 5×1 | 165 |

**Table 4. Network training parameter.**

| Parameters | Parameter Settings |
|---|---|
| Loss | Categorical_crossentropy |
| Optimizer | Adam |
| Metrics | Accuracy |
| Batch_size | 64 |
| Epochs | 100 |
| Activation function(CNN) | ReLU |
| Activation function(LSTM) | Sigmoid |

Using the confusion matrix, we can obtain four commonly used indicators of evaluation models, TP, FN, FP, and TN, where TP means that positive class is predicted to be positive class, FN means that positive class is predicted to be negative class, FP means that negative class is predicted to be positive class, and TN means that negative class is predicted to be

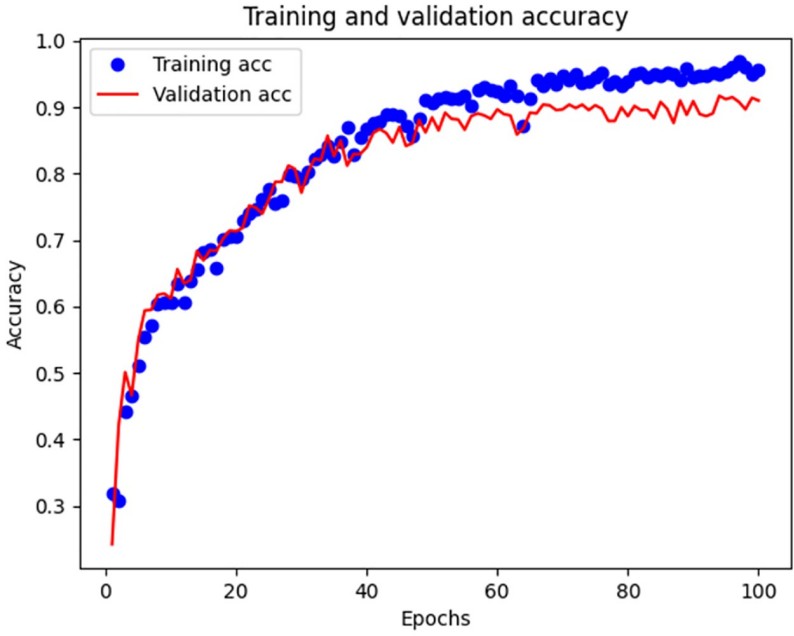

**Fig 6. Variation of accuracy.**

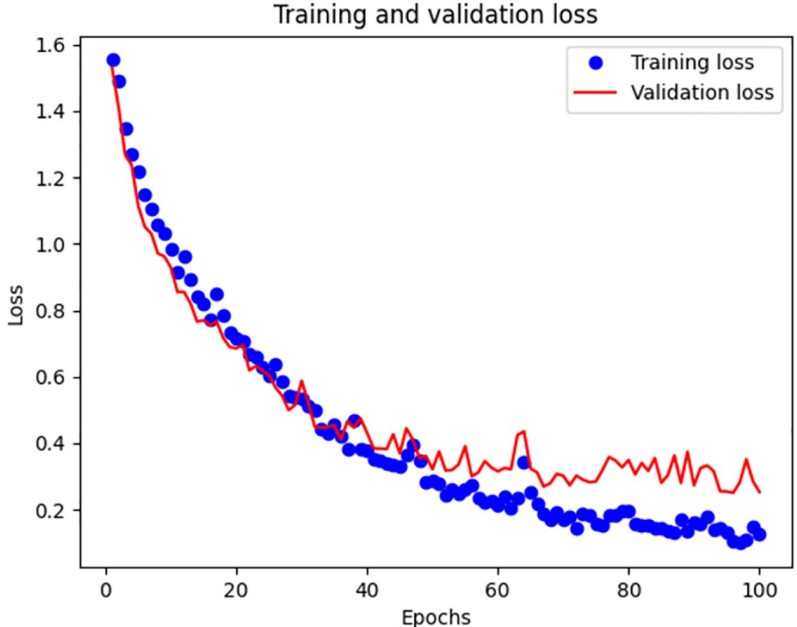

**Fig 7. Variation of loss.**

**Table 5. Comparison of three kinds of network recognition results.**

| Network | Accuracy of training set | Accuracy of test set |
|---|---|---|
| LSTM | 82.27% | 76.10% |
| 1D-CNN | 85.98% | 84.18% |
| Joint Network | 96.73% | 92.14% |

negative class. Therefore, we can calculate the precision and recall of the model. The calculation formula is as follows.

$$precision = \frac{TP}{TP + FP} \qquad (11)$$

$$recall = \frac{TP}{TP + FN} \qquad (12)$$

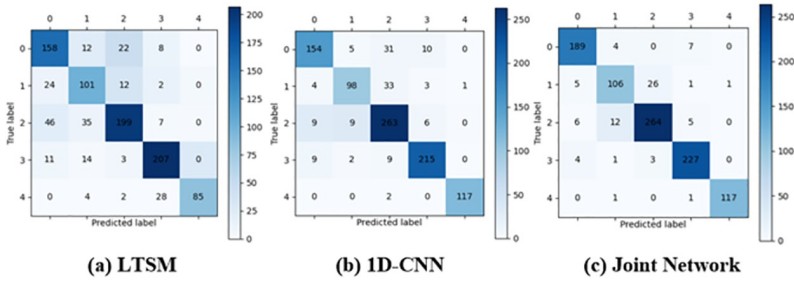

**Fig 8. Confusion matrices for three networks.** (a) LTSM; (b) 1D-CNN; (c) Joint Network.

**Table 6. Various types of recognition.**

| Network | Accuracy of test set | | | | |
|---|---|---|---|---|---|
| | A | B | C | D | E |
| LSTM | 79.00% | 72.66% | 69.34% | 88.09% | 71.43% |
| 1D-CNN | 77.00% | 70.50% | 91.63% | 91.48% | 98.32% |
| Joint Network | 94.50% | 76.26% | 91.99% | 96.60% | 98.32% |

$$F1 = \frac{2 \cdot precision \cdot recall}{precision + recall} \tag{13}$$

For each category, the precision, recall and F1 score were calculated. The results are shown in Figs 9–11 respectively.

The figure reveals that the joint network proposed in this study performs better than the traditional single network in all aspects, especially in the F1 Score. As the F1 score considers the precision and recall rate, it is more comprehensive to evaluate the network with the F1 score. We can see that the target recognition score of the joint network is higher than that of the traditional single network.

To ensure repeatability, we conducted 30 training sessions for the three network models, and compared the network recognition accuracy after 30 training sessions. The 30 training results for the three networks are shown in Fig 12.

By conducting 30 experiments, we can see that the joint network proposed in this paper has absolute advantages in all types of recognition results, and the overall recognition effect of 30 times is better than that of a single LSTM network and 1D-CNN.

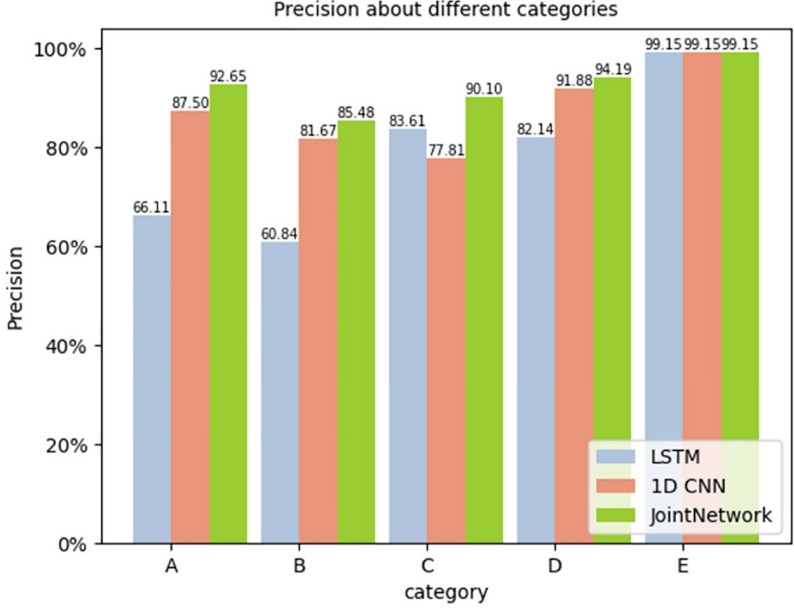

**Fig 9. Precision about different categories.**

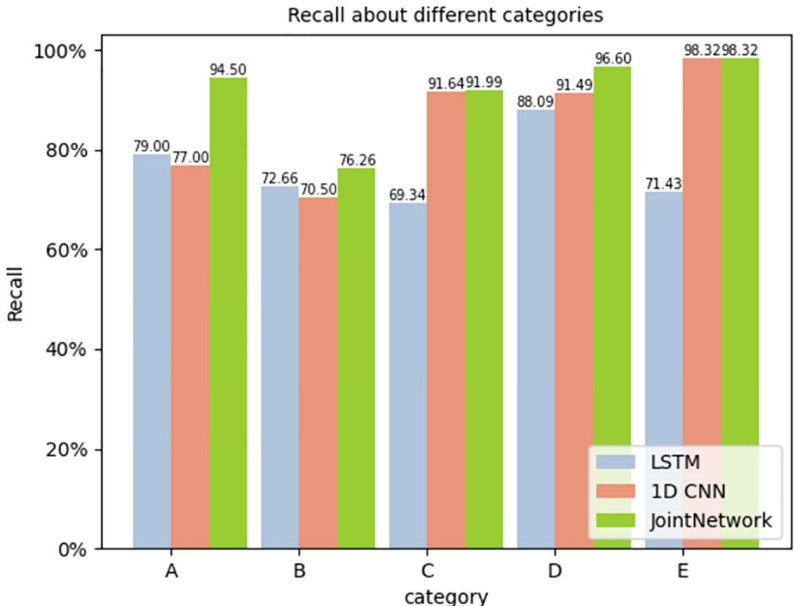

**Fig 10. Recall about different categories.**

However, we can also see that the network does not have a very good recognition effect for Type B, and this is presumed to be caused by the insufficient training of the network owing to the small number of type B samples. Class E, with the same small number of samples is environmental noise, which is clearly differentiated from other categories; thus, the recognition effect is good.

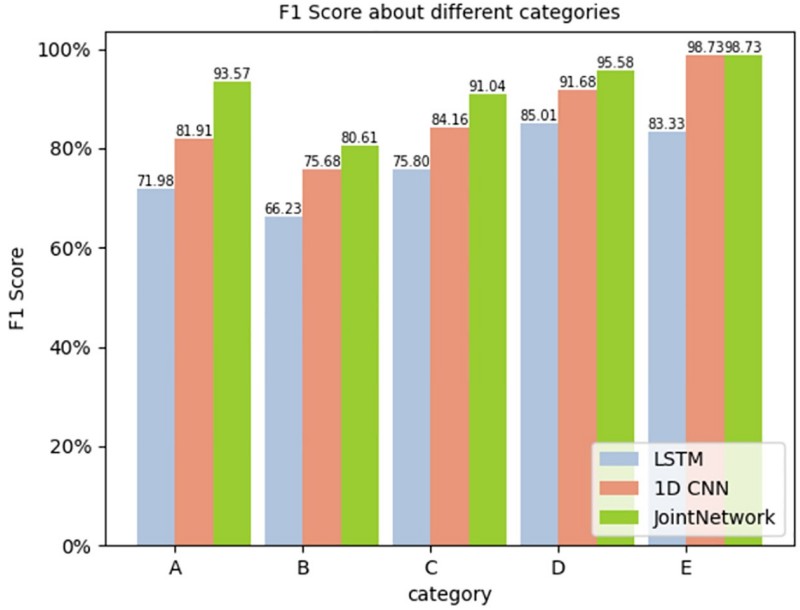

**Fig 11. F1 Score about different categories.**

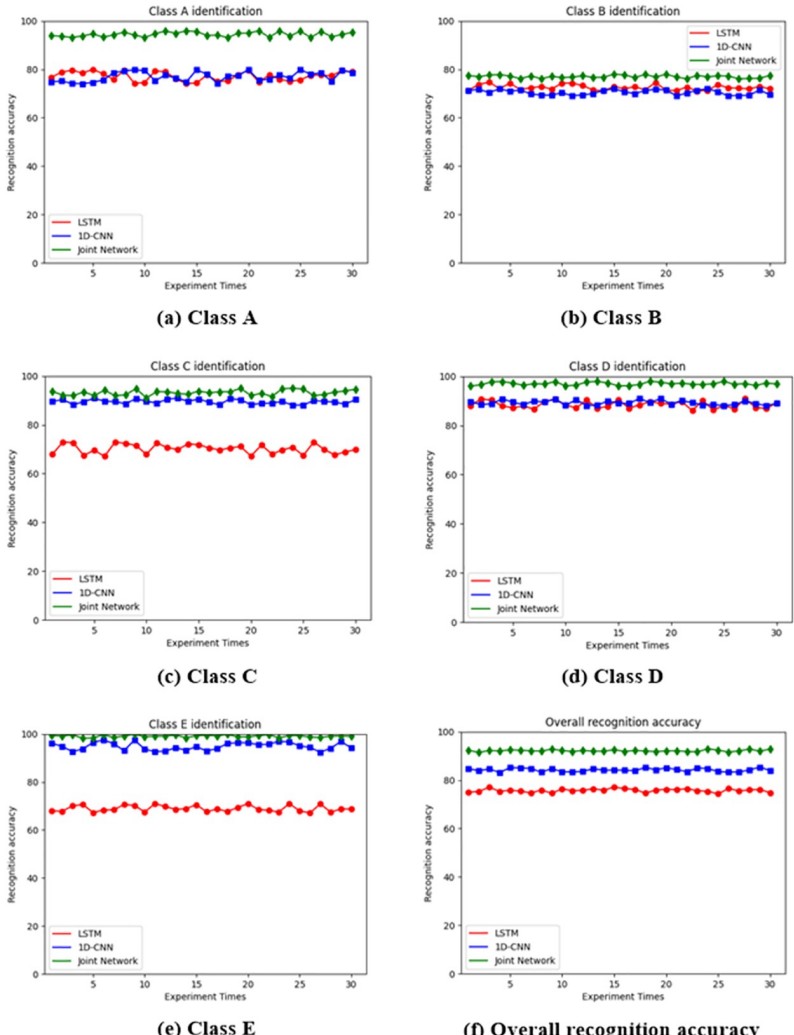

**Fig 12. Comparison results of the recognition accuracy of the three networks.** (a) Class A; (b) Class B; (c) Class C; (d) Class D; (e) Class E; (f) Overall recognition accuracy.

By conducting 30 experiments, we found that the performance of the joint network was robust, which inspired us to use the joint network for underwater acoustic target recognition in the future.

## 5. Conclusion

In this study, a new network structure combining a 1D CNN and LSTM network is proposed and applied to underwater acoustic target recognition. The joint network can combine the advantages of the two neural networks to extract features from input data more comprehensively.

The experimental results using the ShipsEar underwater vessel dataset show that the proposed joint network has a higher recognition rate than traditional neural networks. Compared with 1D CNN and LSTM networks, the joint neural network has higher accuracy, precision, recall and F1 score. The network also has a simple structure, fewer parameters and shorter

training time. This provides a new development direction for underwater acoustic target recognition methods.

The limitation of this study is that only one dataset is used in experimentation. Both the training and test sets originate from the ShipsEar dataset, thus the performance of this network has not been verified in an actual marine environment. Our research direction is to expand the data set, collect more measured ship noise acoustic signals, optimize network parameters by increasing the number of data sets and continuous training, so as to enhance the universality of the network.

## Author Contributions

**Methodology:** Xing Cheng Han.

**Validation:** Liming Wang.

**Visualization:** Yunjiao Bai.

**Writing – original draft:** Chenxi Ren.

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
