## [Decision Letter · Decision Letter 0]

27 Jan 2022

PONE-D-21-38631Underwater Acoustic Target Recognition Method Based on Joint Neural NetworkPLOS ONE

Dear Dr. Han,

Thank you for submitting your manuscript to PLOS ONE. After careful consideration, we feel that it has merit but does not fully meet PLOS ONE’s publication criteria as it currently stands. Therefore, we invite you to submit a revised version of the manuscript that addresses the points raised during the review process.

We look forward to receiving your revised manuscript.

Kind regards,

Stavros Ntalampiras

Academic Editor

PLOS ONE

Journal Requirements:

1.** **When submitting your revision, we need you to address these additional requirements.

Reviewers' comments:

Reviewer's Responses to Questions

**Comments to the Author**

1. Is the manuscript technically sound, and do the data support the conclusions?

Reviewer #1: Partly

Reviewer #2: Yes

2. Has the statistical analysis been performed appropriately and rigorously? 

Reviewer #1: Yes

Reviewer #2: No

3. Have the authors made all data underlying the findings in their manuscript fully available?

Reviewer #1: Yes

Reviewer #2: No

4. Is the manuscript presented in an intelligible fashion and written in standard English?

Reviewer #1: No

Reviewer #2: No

5. Review Comments to the Author

Reviewer #1: The authors present an underwater target classification approach combining CNN and LSTM. The authors have compared the results of the joint network with that of the individual networks and the proposed architecture yields good classification results on ShipsEar dataset.

While the study appears to be sound, there are some significant issues that need to be addressed, some of which are listed below.

1) The language is unclear in many places, making it difficult to understand. Moreover, there are several grammatical mistakes. Many sentences are poorly constructed such as the first line of Abstract, a paragraph in Section 2.1 starting with “Due to extract the characteristics……...average pooling”, a paragraph in Section 3.2 starting with “The first is to extract……..eigenvector of (128,1)”, the first paragraph of Conclusion etc. Such mistakes can be noticed throughout the manuscript. So, it is highly recommended that the authors perform a thorough proofread before submitting the manuscript.

2) In various places, the authors claim that the joint network is proposed for the first time. There are several articles that utilize combined CNN and LSTM in underwater target recognition; the only difference here is that instead of 2D CNN, the authors make use of 1D CNN. So, it will be better to avoid such writings, or else, they should specifically emphasize the novelty of the current work by focusing on 1D CNN.

3) The authors need to significantly improve the technicality of the manuscript. The contents related to underwater target recognition and network architecture should be more technical. In many places, descriptions are provided in a casual manner. It is recommended to refer to good articles published in these domains to get a good technical background.

1. Introduction

1) The authors need to properly establish why they have come up with such an architecture. In this context, they must elaborate on the shortcomings/drawbacks of the existing models in underwater target recognition, such as joint architectures of 2D CNN and LSTM as well as 1D CNN and LSTM employing filterbank learning from raw acoustic inputs. The Introduction section need to be reframed accordingly to include a concise literature survey on these topics and a proper justification of the proposed architecture.

2) It seems a good practice to provide citations to works or use et al (with reference) rather than mentioning the full affiliations of the authors (as can be seen in the third and fourth paragraphs).

3) Correct the grammatical mistakes in various places.

2. Recognition principle

1) It will be better to reduce the theoretical part of CNN and LSTM; instead the authors may concentrate more on why CNN and LSTM have been chosen in the context of the current study. Also, it is not of much relevance to provide Fig. 1 & Fig. 2, as they are generic and well known.

2) The activation function in Fig. 1 is misleading. If the authors intended to show ReLU, draw it appropriately.

3) It is recommended to improve the drawing quality of Fig. 2, especially the arrangement of bottom arrows.

4) Many terms in the equation part of LSTM appear to be missing.

5) Correct the grammatical errors.

3. Experimental data

1) As seen in its original paper, the name of the dataset is ‘ShipsEar’. The authors have mentioned it as ‘SHIPSEAR’. Change it accordingly in all places.

2) Use capital letters for starting a title word such as ‘Category’ and ‘Ship types’ in Table 1.

3) In the Data processing section, the authors need to justify why they have used the mentioned features. Also, the formula for the extraction of these features must be provided or at least they must be cited.

4) Have the authors experimented with a single feature like Mel spectrogram alone or MFCC alone? If so, what is the trend of accuracy? How the authors have come up with 5 features? Is there any feature selection method employed?

5) It is seen that the authors have compressed the Mel spectrogram to obtain 1D data to feed to a 1D convolution layer. Have the authors experimented with 2D Mel spectrogram along with 2D CNN? If so, compare the results to justify why the proposed architecture is better.

6) What is the significance of the audio serial number in Table 2? Is there any specific reason for showing it in table?

7) In the second paragraph of Section 3.3, the authors have mentioned “The characteristics of ship…...current moment”. What is meant by that? Use proper technical words to construct such contexts.

8) The direction of writing in Fig. 4 is non-standard. It is preferable to reverse the direction for improved readability (refer to such diagrams in other articles).

9) It is recommended to provide 1 or 2 sentences describing the overall structure of the network, such as the number of convolution layers, max pooling, LSTM and dense layers. The details of individual layer can however be provided in a table, as in Table 3.

10) Nothing is mentioned about the activation function used.

11) Table 3 is an exact copy of the model summary obtained with that particular deep learning framework/library. It is recommended to modify it while including in a technical paper, such as replacing ‘None’ by the corresponding batch size, naming each layer appropriately etc.

12) Correct the grammatical mistakes here and there.

4. Experimental results

1) What is the validation data used in Fig. 5 and Fig. 6? Is it the test data itself or is it derived from the training set? It is always recommended to use separate training, validation and test sets. Provide good captions for the figures such as using ‘variation’ instead of ‘change’.

2) The authors should modify the caption for Fig. 7.

3) Are figures 8, 9 and 10 actually needed? Sufficient information is already available in Fig. 7. Hence, Fig. 8-10 appear to be redundant and space consuming. The authors may provide other relevant performance metrics instead of these.

4) It is better to use P-R curves instead of Table 7 in order to facilitate easy and appealing comparison of the three networks.

5) Use appropriate technical words in the paragraph starting with “In order to prevent the experimental...”. The authors may use “to ensure repeatability” like that.

6) After Fig. 11, the authors have mentioned that the joint network is performing well. They must also write an inference on why some classes have low recognition.

7) Table 8 is absolutely not needed. It the authors want to show the variations among different trials for the three networks, they may use a box plot.

8) Correct the sentence construction and grammatical errors.

5. Conclusion

1) The first sentence is poorly written. Reframe it.

2) Modify conclusion to include certain promising values of the experimental results.

3) It is mentioned that the dataset used is single. What is meant by that? Reframe it for better understanding.

Reviewer #2: 1. The authors are suggested to carefully explain the values of parameters related with neural networks used in simulation;

2. The authors discuss the main structure of neural networks used in this paper, such as CNN and LSTM. However, these two structures are the main popular neural networks widely used in different applications. Therefore, the authors are suggested to highlight the main contributions of this paper and differences between the current work and some published works.

3. The English wording and grammar should be carefully polished to mitigate misunderstanding.

6. PLOS authors have the option to publish the peer review history of their article (what does this mean?). If published, this will include your full peer review and any attached files.

Reviewer #1: No

Reviewer #2: No

---

## [Author Response · Author response to Decision Letter 0]

10 Feb 2022

The authors are very grateful to the reviewers and editor for their valuable comments. We have responsed the comments according to the suggestions.However due to some formulas, pictures and tables in the responses cannot be reflected here, so we wrote a document about the responses to the comments and uploaded it to the system. The document name is :“Response to Reviewers” .So reviewers and editor can view this document,thank you!

---

## [Decision Letter · Decision Letter 1]

28 Feb 2022

PONE-D-21-38631R1Underwater acoustic target recognition method based on a joint neural NetworkPLOS ONE

Dear Dr. Han,

Thank you for submitting your manuscript to PLOS ONE. After careful consideration, we feel that it has merit but does not fully meet PLOS ONE’s publication criteria as it currently stands. Therefore, we invite you to submit a revised version of the manuscript that addresses the points raised during the review process.

We look forward to receiving your revised manuscript.

Kind regards,

Stavros Ntalampiras

Academic Editor

PLOS ONE

Journal Requirements:

Reviewers' comments:

Reviewer's Responses to Questions

**Comments to the Author**

1. If the authors have adequately addressed your comments raised in a previous round of review and you feel that this manuscript is now acceptable for publication, you may indicate that here to bypass the “Comments to the Author” section, enter your conflict of interest statement in the “Confidential to Editor” section, and submit your "Accept" recommendation.

Reviewer #1: (No Response)

Reviewer #2: (No Response)

2. Is the manuscript technically sound, and do the data support the conclusions?

Reviewer #1: Partly

Reviewer #2: (No Response)

3. Has the statistical analysis been performed appropriately and rigorously? 

Reviewer #1: Yes

Reviewer #2: (No Response)

4. Have the authors made all data underlying the findings in their manuscript fully available?

Reviewer #1: Yes

Reviewer #2: (No Response)

5. Is the manuscript presented in an intelligible fashion and written in standard English?

Reviewer #1: No

Reviewer #2: (No Response)

6. Review Comments to the Author

Reviewer #1: The authors have attempted to incorporate the suggestions provided by the reviewers and have tried to answer various queries raised by the reviewers, but thorough proofreading has not been performed yet. Grammatical and spelling mistakes can still be noticed in various places, and in many places where modifications have been made, errors are prominent. This will significantly affect the quality of the manuscript. Some of the suggestions are as follows.

1) For tables 1 to 5, it will be visually more appealing to use the same format used in the originally submitted manuscript, i.e. use borders only for title and at the bottom than using borders everywhere.

2) Some paragraphs have only one or two lines. Merge paragraphs that are too short.

Abstract:

1) It is more common to use ‘indices’ instead of ‘indexes’.

Introduction:

1) Paragraph 4: In the second line, the continuity for LSTM has been lost; Reframe it.

2) Paragraph 5 & 6: Grammatical errors- ‘seem’, ‘reduce’ and ‘build’ are used instead of ‘seems’, ‘reduces’ and ‘built’ respectively.

Recognition Principle:

Convolutional neural network:

1) Technical mistake in the line: “Finally, the predicted category labels can be obtained through the convolution layers”

2) Paragraph 3: Comma should be provided after respectively; instead it is provided before.

Long short-term memory:

1) Paragraph 3: ‘Forgetting gate’ is provided instead of ‘Forget gate’

2) Incorrect commas between Wi & Wc and between bi & bc, after equation 8.

Data processing:

1) Paragraph 4: Spelling mistake - ‘calcultated’

2) The authors need not elaborate much on audio related theories behind spectral contrast and tonnetz. There is no need to mention the librosa function used to calculate tonnetz; instead provide equations to extract these features, if available. The use of audio everywhere does not seem a good practice in an underwater target recognition paper, since an audio signal is mostly associated with human hearing and spans frequencies from 20 Hz - 20 kHz, which is not the scenario of a ship signal. The authors may instead mention that they have used features that are mostly auditory-inspired and in other places where they have mentioned audio signals, they may either use acoustic signals or target signals (even in Fig 3.)

3) Paragraph 9 : Grammatical mistake in the line “each audio data will obtain a (192,1) feature vector”

4) It is suggested to change the spelling of T-SNE to t-SNE in multiple places.

5) Wrong title for Figure 4 wrong - t-SNE embedding of digits?

6) Provide legends in Figure 4 to understand the various categories.

Network Construction:

1) Paragraph 3: Why 'DENSE' is provided in capital letters?

Training results:

1) Paragraph 3: Incorrect sentence construction - “To verify whether the performance of the combined network proposed in this study…….”

2) Redundancy in the word ‘category’ just before Fig 9.: “For each category, the precision, recall and F1 score for this category…”

3) Why it is mentioned “The result of the precision…” in Fig 9, “The result of the recall…” in Fig. 10 and “The result of F1 score…” in Fig. 11? Precision, recall and F1-score themselves are results. Simply write Precision, Recall and F1-score for the figure titles & captions. Provide a label such as ‘category’ for x-axis.

4) The caption for Fig. 11 is wrongly provided.

5) Grammatical mistake in the paragraph after Fig. 11: “evaluating the network in terms of the F1 score it is more comprehensive”

Conclusion:

1) It was suggested to include certain promising values of experimental results, but it is still not incorporated. For this, the authors may write a sentence with the values of accuracy, P, R and F1 of the proposed joint network, compared against LSTM and 1D CNN.

2) Paragraph 3: Grammatical mistake - “performance of this network was not be verified in an actual marine environment…”

3) Last sentence is too long and incorrectly constructed. Split the sentence instead of using so much ‘and’.

Reference

1) Reference [30] still shows “SHIPSEAR” instead of “ShipsEar”.

Reviewer #2: The authors have already revised the draft based on the reviewers' comments. I suggest to accept this paper after checking wording and format.

7. PLOS authors have the option to publish the peer review history of their article (what does this mean?). If published, this will include your full peer review and any attached files.

Reviewer #1: No

Reviewer #2: No

---

## [Author Response · Author response to Decision Letter 1]

3 Mar 2022

The authors are very grateful to the reviewers and editor for their valuable comments. We have responsed the comments and upload the responses as a document with the name:“Response to Reviewers”.Please see this document "Response to Reviewers.docx", thank you!

---

## [Decision Letter · Decision Letter 2]

21 Mar 2022

Underwater acoustic target recognition method based on a joint neural Network

PONE-D-21-38631R2

Dear Dr. Han,

We’re pleased to inform you that your manuscript has been judged scientifically suitable for publication and will be formally accepted for publication once it meets all outstanding technical requirements.

Please make sure to implement the final minor comments made by the Reviewers.

Kind regards,

Stavros Ntalampiras

Academic Editor

PLOS ONE

Additional Editor Comments (optional):

Reviewers' comments:

Reviewer's Responses to Questions

**Comments to the Author**

1. If the authors have adequately addressed your comments raised in a previous round of review and you feel that this manuscript is now acceptable for publication, you may indicate that here to bypass the “Comments to the Author” section, enter your conflict of interest statement in the “Confidential to Editor” section, and submit your "Accept" recommendation.

Reviewer #1: (No Response)

Reviewer #2: All comments have been addressed

2. Is the manuscript technically sound, and do the data support the conclusions?

Reviewer #1: Partly

Reviewer #2: Yes

3. Has the statistical analysis been performed appropriately and rigorously? 

Reviewer #1: Yes

Reviewer #2: Yes

4. Have the authors made all data underlying the findings in their manuscript fully available?

Reviewer #1: Yes

Reviewer #2: Yes

5. Is the manuscript presented in an intelligible fashion and written in standard English?

Reviewer #1: No

Reviewer #2: Yes

6. Review Comments to the Author

Reviewer #1: The authors have tried to incorporate the comments/revisions suggested by the reviewer. However, in some places, the authors have misunderstood what the reviewer really meant. Also, some minor grammatical mistakes still exist in corrected sections. The authors are suggested to correct those in the final manuscript.

1) Correct the 4th paragraph of Introduction section as follows.

"The long short-term memory (LSTM) architecture is suitable for processing and forecasting events with long intervals in time series. The analysis of ship-radiated noise depends largely on local time-frequency information and time-series related information; therefore, LSTM can be utilized for underwaster acoustic target recognition [14-17]."

2) Change the last line of 1st paragraph of section 2.1 as "Finally, the predicted category labels can be obtained through the fully connected output layer."

3) Reframe the two lines in the 3rd paragraph of section 2.2 as,

i) "The update calculation of forget gate implementation is as follows:"

ii) "The forget gate reads...............................completely dropped."

4) Combine the 2nd and 3rd lines of 1st paragraph of section 3.1 as follows.

"This dataset consists of 90 acoustic records of 11 types of ships and environmental noise within 15 seconds – 10 minutes."

5) Change the last line of 1st paragraph of section 3.2 as follows.

"therefore, the dataset of 90 acoustic signals needs to be preprocessed."

6) Change the 9th paragraph of section 3.2 as,

"After the five features are extracted, the feature vectors obtained are fused, and for each acoustic signal, a feature vector with a dimension of (192,1) is provided as the input of the network."

7) Change the paragraph before Table 2 as,

"To make it easier for other researchers to use the ShipsEar dataset, each acoustic signal piece in the dataset is assigned a number, and the serial number used is indicated in Table 2."

8) Change the title in Table 2 from "Acoustic signals serial number" to "Acoustic signal serial number"

9) Change the first line of 2nd paragraph of Conclusion as follows.

"The experimental results using the ShipsEar underwater vessel dataset show that the proposed joint network has a higher recognition rate than traditional neural networks. Compared with 1D CNN and LSTM networks, the joint neural network has higher accuracy, precision, recall and F1 score.The network also has a simple structure, fewer parameters and shorter training time."

Reviewer #2: The authors have carefully addressed all comments. I suggest to accept this paper after checking format and wording.

7. PLOS authors have the option to publish the peer review history of their article (what does this mean?). If published, this will include your full peer review and any attached files.

Reviewer #1: No

Reviewer #2: No

---

## [Editor Report · Acceptance letter]

8 Apr 2022

PONE-D-21-38631R2 

Underwater acoustic target recognition method based on a joint neural Network 

Dear Dr. Han:

I'm pleased to inform you that your manuscript has been deemed suitable for publication in PLOS ONE. Congratulations! Your manuscript is now with our production department. 

Kind regards, 

on behalf of

Prof. Stavros Ntalampiras 

Academic Editor

PLOS ONE